# Low-Level Clostridial Spores’ Milk to Limit the Onset of Late Blowing Defect in Lysozyme-Free, Grana-Type Cheese

**DOI:** 10.3390/foods12091880

**Published:** 2023-05-02

**Authors:** Domenico Carminati, Barbara Bonvini, Salvatore Francolino, Roberta Ghiglietti, Francesco Locci, Flavio Tidona, Monica Mariut, Fabio Abeni, Miriam Zago, Giorgio Giraffa

**Affiliations:** Council for Agricultural Research and Economics, Research Centre for Animal Production and Aquaculture (CREA-ZA), Via Lombardo 11, 26900 Lodi, Italy; domenico.carminati@crea.gov.it (D.C.); barbara.bonvini@crea.gov.it (B.B.); salvatore.francolino@crea.gov.it (S.F.); roberta.ghiglietti@crea.gov.it (R.G.); francesco.locci@crea.gov.it (F.L.); flavio.tidona@crea.gov.it (F.T.); monica.mariut@crea.gov.it (M.M.); fabiopalmiro.abeni@crea.gov.it (F.A.); miriam.zago@crea.gov.it (M.Z.)

**Keywords:** Grana Padano cheese, cheese ripening, cheese spoilage, cheese microbiota, anaerobic spore-forming bacteria, milk quality

## Abstract

The growth of clostridial spores during ripening leads to late blowing (LB), which is the main cause of spoilage in Grana Padano Protected Designation of Origin (PDO) cheese and other hard, long-ripened cheeses such as Provolone, Comté, and similar cheeses. This study aimed to verify the cause–effect relationship between the level of clostridial butyric spores (BCS) in milk and the onset of the LB defect. To this end, experimental Grana-type cheeses were produced without lysozyme, using bulk milk with different average BCS content. The vat milk from the so-called “virtuous” farms (L1) contained average levels of BCS of 1.93 ± 0.61 log most probable number (MPN) L^−1^, while the vat milk from farms with the highest load of spores (L2), were in the order of 2.99 ± 0.69 log MPN L^−1^. Cheeses after seven months of ripening evidenced a strong connection between BCS level in vat milk and the occurrence of LB defect. In L2 cheeses, which showed an average BCS content of 3.53 ± 1.44 log MPN g^−1^ (range 1.36–5.04 log MPN g^−1^), significantly higher than that found in L1 cheeses (*p* < 0.01), the defect of LB was always present, with *Clostridium tyrobutyricum* as the only clostridial species identified by species-specific PCR from MPN-positive samples. The L1 cheeses produced in the cold season (C-L1) were free of defects whereas those produced in the warm season (W-L1) showed textural defects, such as slits and cracks, rather than irregular eyes. A further analysis of the data, considering the subset of the cheesemaking trials (W-L1 and W-L2), carried out in the warm season, confirmed the presence of a climate effect that, often in addition to the BCS load in the respective bulk milks (L1 vs. L2), may contribute to explain the significant differences in the chemical composition and some technological parameters between the two series of cheeses. Metagenomic analysis showed that it is not the overall structure of the microbial community that differentiates L1 from L2 cheeses but rather the relative distribution of the species between them. The results of our trials on experimental cheeses suggest that a low-level BCS in vat milk (<200 L^−1^) could prevent, or limit, the onset of LB in Grana-type and similar cheeses produced without lysozyme.

## 1. Introduction

Butyric acid fermentation is still a major cause of spoilage in semi-hard and hard cheeses, leading to the generation of the late blowing (LB) defect, caused by the outgrowth of clostridial spores during ripening. Butyric clostridia are strict anaerobic microorganisms that, by consuming lactate or the residual lactose, carry out the butyric fermentation with accumulation of butyric acid, acetic acid, and high quantities of gases (CO_2_ and H_2_) that cause an irregular eye formation (which, in the most serious cases, can also visibly deform the cheese), slits, and off-flavors during ripening [1,2]. Late blowing, which is generally caused by a restricted number of butyric clostridial species such as *Clostridium tyrobutyricum* (the main blowing agent and the most frequent isolated species), *Clostridium sporogenes*, *Clostridium butyricum*, and *Clostridium beijerinckii* [3,4], gives rise to relevant economic losses, especially in cheeses with a high commercial diffusion and consumption such as Grana Padano (GP) PDO cheese. With a production size of around 200,000 tons in 2021 (over 5,200,000 cheeses) and an ever-expanding export (44% of total production; 2,288,000 cheeses in 2021), GP is indeed among the most known, consumed, and appreciated Italian cheeses in the world (https://www.granapadano.it, accessed on 24 April 2023). On the other hand, LB is still a notable drawback in GP production, leading to a great loss in product value. This is mainly related to the use of corn silage as the main forage source for feeding dairy cows that produce milk for GP. Corn silage and other ensiled forages are recognized as the main contamination source of BCS responsible for LB defect in hard and semi-hard long-ripened cheese production. Through a chain starting from the silage to the total mixed ration (TMR), the feces, and the cow udder, the milk contamination finally takes place [5]. The extent of this contamination varies across seasons and both the feed’s microbiological quality and the management practices at the farm may affect the level of BCS in milk for GP cheese production.

To date, lysozyme is still the most applied and successful option for limiting the onset of LB defect in GP and similar hard cheeses. Consumer needs for clean label products and the legal requirement to indicate lysozyme as a potential allergen, due to its extraction from egg, are driving the industry to look for alternative solutions (e.g., mechanical removal using bactofugation, application of protective cultures) to counteract the onset of LB [6]. While recognizing the whey starter culture and a correct cheese processing a fundamental role in limiting the germination and development of BCS, a GP cheese obtained from milk with high levels of anaerobic spore-formers will most likely show LB problems. To avoid the use of lysozyme, controlling the number of spore-forming bacteria in milk therefore remains one of the last and best options [7,8]. This study aimed to verify the cause–effect relationship between the load of BCS in milk and the onset of the LB defect. To this end, experimental Grana-type cheeses manufactured without lysozyme and using bulk milk with different average BCS content were produced. In this regard, two farms with low titer (named L1, <200 spores L^−1^) and two with high titer (named L2, >600 spores L^−1^) were identified. The cheeses were chemically analyzed, the number of BCS and other spore-formers quantified, and the frequency of the appearance of LB symptoms during ripening was recorded. At the end of the ripening, a metagenomic analysis to compare the bacterial diversity between the two series of cheeses was carried out. Although a low number of spores in milk may contribute to obtaining LB-free Grana-type hard cheeses, our results leave the question open on the role of technological and chemical features to control the onset of the LB defect, especially in cheeses produced during the hot season.

## 2. Materials and Methods

### 2.1. Cheese Manufacturing and Experimental Design

Raw cow milk for experimental cheesemaking trials was collected from four farms located in the province of Cremona (Lombardy region, northern Italy). Farms belonged to a dairy cooperative that confers raw milk for manufacturing of Grana Padano PDO cheese. The farms were selected within a first study on the effects of different BCS contamination chains from forage to milk in 108 herds [5]. Based on the yearly mean BCS titer of spores, two farms with low titer (named L1, <200 spores L^−1^) and two with high titer (named L2, >600 spores L^−1^) were identified. To minimize the composition variability between the different cheesemaking trials, the raw milk was partially skimmed using natural creaming to reach a fat/protein ratio of ∼0.80. Twelve cheesemaking trials (six with milk L1 and six with milk L2) were carried out at the dairy pilot plant CREA-ZA (Lodi, Italy) following a Grana-type cheesemaking process [9], in compliance with the production specifications of Grana Padano PDO (www.granapadano.it, accessed on 24 March 2023), but without adding lysozyme. Twelve cheese wheels with L1 and twelve with L2 milk were produced, for a total of 24 wheels. The cheeses were produced once a week in a period between April and October, alternating each time the use of 1100 L of L1 or L2 milk, respectively. For each cheese making, two bell-shaped copper vats, containing 550 L/each of milk, were used. One cheese wheel/vat was obtained.

### 2.2. Sampling

For each cheesemaking trial, samples of vat milk (after natural creaming), curd (at first turning after 3 h of molding), and cheese (after 10 months of ripening) were collected. All samples were transported to the laboratory under refrigeration (4 °C) and analyzed no later than 2 h from collection. For cheeses, a slice representative of the wheel radius, i.e., from the center to the rind (20 × 5 cm; 1 kg) was sampled. pH, salt, and organic acids determinations (see below) were carried out on a sample representative of the whole slice, and on subsamples taken from the inner (center of the wheel) and the outer part (3 cm below the rind) of the slice. The cheeses were evaluated as affected or not affected by LB by visual inspection on structural defects (presence of cracks, swellings).

### 2.3. Chemical Analysis

Fat and protein content of milk were determined using a FT-IR Milkoscan FT2 (Foss, Padova, Italy). The pH of both milk and cheeses was measured using a portable pH-meter (Portavo-907, Knick, Germany). Acidity of milk and whey starters was evaluated by titration with 0.25 N NaOH and expressed as °SH/50 mL. The fat, protein, total solids, ash, and NaCl content of the samples of curd and/or cheese were determined according to IDF Standards 17A, 27, 4, 20-1, and 238 [10,11,12,13,14], respectively. The lactic, butyric, propionic, acetic, and pyroglutamic acid contents of ripened cheeses were determined using HPLC according to Bouzas et al. [15]. HPLC analysis was carried out under isocratic conditions at 0.6 mL min^−1^ and 65 °C using a 300 × 7.8 mm cation exchange column (Aminex HPX-87H). Samples were prepared as follows: 25 mL of 0.009 N H_2_SO_4_ was added to 5 g of ground cheese and mixed with a magnetic stirrer for 30 min. The mixture was centrifuged at 5000× *g* for 10 min and the supernatant was filtered through a 0.45 μm cellulose acetate membrane syringe filter (Bio-Rad Laboratories, Richmond, CA, USA).

### 2.4. Microbiological Analysis

Cheese samples were first shredded for 1 min in a sterile blender, then 10 g was suspended in sterile 2% trisodium citrate solution (1:10 *w*:*v*) and homogenized for 2 min in a Stomacher 400 Circulator blender (Seward Ltd., London, UK). The vat milk samples were shaken 10 times before analysis. Tenfold dilutions were performed in sterile quarter-strength Ringer solution (Thermo Scientific Oxoid, Basingstoke, UK). Cheese samples were analyzed for enumeration of anaerobic spore-forming bacteria (ANSB), butyric clostridia spores (BCS), aerobic spore-forming bacteria (ASB), total mesophilic bacteria (TMB), and propionibacteria (PB). Vat milk samples were only analyzed for ANSB and BCS spore content. The number of ANSB spores was determined using a most probable number (MPN) method with a 3 × 3 scheme, three 10-fold dilutions and three tubes for each dilution. One-milliliter aliquots of decimal dilutions were inoculated into 3 tubes each containing a culture medium (9 mL) based on reconstituted skimmed milk (10% *w*/*v*) supplemented with a selective medium according to Zucali et al. [16]. The inoculated tubes were then sealed with 2 mL of melted 1:5 paraffin:vaseline blend (Sacco System srl, Cadorago, Italy), and heated at 80 °C for 10 min to inactivate vegetative cells and to trigger the germination of spores. Tubes were incubated at 37 °C for 7 d. A positive result (i.e., gas production) was shown by the completely lifted paraffin plug. The number combination of gas-positive tubes in the last three serial dilutions was used to evaluate the number of spores. MPN counts and its parameters were estimated using a freely available Excel spreadsheet were developed by Jarvis et al. [17] and the results were expressed according to the starting matrix amount (spore log MPN g^−1^ or L^−1^).

The presence of BCS in gas-positive tubes from ANSB enumeration was detected using a multiplex PCR assay for butyric clostridia species (*Cl. tyrobutyricum*, *Cl. butyricum*, *Cl. beijerinckii*, and *Cl. sporogenes*) targeting *enr* (2-enoate reductase), *hydA* (hydrogenase), *nifH* (nitrogenase iron protein), and *colA* (collagenase), respectively, from bacterial DNA extracted from each tube, according to Cremonesi et al. [18]. Each sample (400 μL) was placed in a 2 mL tube containing one stainless steel bead (5 mm diameter). Lysis buffer (reagent AL, 400 μL) and antifoam (reagent DX, 1 µL) were added to each tube. Samples were treated twice in the automated homogenizer Tissue Lyser II (Qiagen, Milan, Italy) for 5 min at 30 Hz. After centrifugation at 3000× *g* for 2 min, 400 µL of supernatant was collected and DNA was extracted using the QIAamp 96 QIAcube HT kit (Qiagen), according to manufacturer instructions. PCR products were uploaded in the QIAxcel instrument (Qiagen) for separation of fragments by electrophoresis under controlled conditions as per manufacturer’s instructions. QIAxcel DNA Screening Gel Cartridge was used for the study. The QX Alignment Marker (15 bp–1 kb) and the QX DNA Size Marker (500–800 bp) were included in every run. Results, displayed in both electropherogram and gel image formats, were analyzed using QIAxcel ScreenGel software. Based on the PCR results from gas-positive tubes, the MPN values were reassessed and the number of BCS spores were deduced. For enumeration of ASB spores, 10-fold dilution of samples were heated at 80 °C for 10 min, plated on Plate Count Milk Agar (PCMA, Thermofisher-Oxoid) and incubated at 37 °C for 48 h. Total mesophilic bacteria (TMB) and propionibacteria (PB) were counted on PCMA at 30 °C for 3 d, and Pal Propiobac agar medium (Laboratoires Standa, France) at 30 °C for 6 d under anaerobic conditions [9], respectively.

### 2.5. Metagenomic Analysis

Metabarcoding analysis was carried out on total DNA extracted from 10 g of the six L1 and the six L2 Grana-type cheese samples treated as described by Zago et al. [19,20]. Total DNA of cheese samples was subjected to metabarcoding analysis at GalSeq laboratories (Bresso, Italy) by sequencing of the variable V3–V4 regions of the 16S rRNA gene using an Illumina HiSeq platform, as described previously [17,18]. Reads were demultiplexed based on Illumina indexing system. Following the QIIME pipelines, the USEARCH algorithm (version 8.1.1756, 32-bit) (https://drive5.com/usearch/, accessed on 2 February 2021) allowed the following steps: chimera filtering; grouping of replicate sequences; sorting sequences per decreasing abundance; identification of the operational taxonomic unit (OTU) with a species-level taxonomic resolution. When the taxonomy assignment did not reach the species level, the genus or family name were reported (19, 20). OTUs < 5 reads were removed. On the resulting OTU table, the alpha (α) diversity (richness and Shannon indexes) and the rarefaction curves were assessed using R software (http://www.r-project.org/index.html, accessed on 26 February 2021) with the “vegan” [21] and “agricolae” [22] packages. Relative abundance for each OTU across the samples was calculated and “subdominant” and “dominant” OTUs were discriminated as 0.1–1% and ≥1% of relative abundance, respectively. Taxonomic analysis of the bacterial communities was performed and visualized by using “reshape2” and “ggplot2” packages, according to Zago et al. [19,20]. 

### 2.6. Statistical Analysis

The effects of starting milk quality and time of milk processing were evaluated, for each considered variable, with the analysis of variance on balanced data, with a two-way model, using R software, also testing their simple interaction. The MPN data of spore counts were converted into log_10_ for statistical analysis. To calculate averages, the values below the detection limit (i.e., 360 spores L^−1^ in milk samples; 3.6 spores g^−1^ in cheese samples) were assigned the lower value of the 95% confidence limits of the MPN detection limit (i.e., 48 spores L^−1^ in milk samples; 0.48 spores g^−1^ in cheese samples). For MPN results above the maximum value, i.e., when all the tubes of the last three inoculated dilutions were gas-positive, the maximum values were assigned.

## 3. Results and Discussion

In this work, the possibility of eliminating the addition of lysozyme to raw milk with a low BCS number, compared to milk naturally rich in spores, was evaluated by way of experimental Grana cheesemaking. The two series of Grana-type cheeses were analyzed during ripening and compared for microbiological and chemical parameters and the possible onset of clostridial LB defect. The choice of the so-called “virtuous” farms, i.e., suppliers of milk with low numbers of BCS, as well as of those with a high number of spores, was based on preliminary information of the average number of BCS, on a yearly basis, of many milk producers in the GP cheese area. 

First, our results confirmed that the vat milk from virtuous farms (L1) contained average levels of BCS of 1.93 ± 0.61 log MPN L^−1^, while the vat milk from farms with the highest load of spores (L2), mostly belonging to *Cl. tyrobutyricum* and *Cl. sporogenes*, were in the order of 2.99 ± 0.69 log MPN L^−1^ (Table 1). This difference, highly significant (*p* < 0.05) between the two series of milks, was the preliminary condition to facilitate the occurrence of the LB defect in Grana-type cheeses produced without the use of lysozyme. Indeed, the threshold value of BCS in milk that can cause the LB defect was indicated between 2.8 and 3.0 log MPN L^−1^, but lower values (i.e., <2.0 log MPN L^−1^) have also been reported [4,15]. Borreani et al. [23] split the spore contamination levels that could influence the onset of LB in hard cheese making process into three classes: <2.3 (low probability), 2.3–3.0 (medium probability), and >3.0 (high probability) log MPN L^−1^, respectively. To note, threshold values may depend on the cheese type being produced and the MPN-based count methods [24]. However, studies on the correlation between BCS numbers in milk and incidence of LB defect in cheese are limited, and information is often based on personnel experience [25].

The analysis of cheeses after at least 7 months of ripening evidenced the strong connection between BCS level in vat milk and the occurrence of LB defect. In L2 cheeses, which showed an average BCS content of 3.53 ± 1.44 log MPN g^−1^ (range 1.36–5.04 log MPN g^−1^), significantly higher than that found in L1 cheeses (*p* < 0.001), the defect of LB was always present and *Cl. tyrobutyricum* was the only clostridial species identified by species-specific PCR from MPN-positive tubes (Table 1).

All L2 cheeses showed slits and cracks, but three of them also presented regular eyes typical of propionic fermentation (Appendix A; cheeses 4, 8, and 10), whose occurrence was confirmed by the propionic acid detected in the inner part of cheese wheels (Appendix A) and, in cheese 8, also by a detectable load of propionibacteria (2.74 log cfu g^−1^) (Appendix A). 

Different results were found for L1 cheeses, depending on the seasonal period in which the cheesemaking trials were carried out. The L1 cheeses produced in the cold season (C-L1) were free of defects whereas those produced in the warm season (W-L1) showed textural defects, such as slits and cracks, rather than irregular eyes (Appendix A: W-L1 cheeses nr 3, 5, 7). In W-L1 cheeses, *Cl. tyrobutyricum* spores and butyric acid in the inner part of the cheese wheel were detected (Table 1; Appendix A). Regarding the other microbial groups, no significant differences were observed between L1 and L2 cheeses (Appendix A), except for overall anaerobic spores (ANSB including BCS), which were significantly higher (*p* < 0.01) in the latter (Table 1).

The onset of LB defect in cheeses is the result of several and often concomitant causes, not always and necessarily related to the number of BCS in the milk [26]. The chemical analysis of vat milk and aged Grana-type cheeses contributed, on the one hand, to explain the LB in mature cheeses and, on the other hand, to confirm that the defect was predominantly caused by butyric acid fermentation. The mean values of the main chemical and physical parameters of the milk before (bulk milk, Table 2) and after creaming (vat milk, Table 3) were different for the L1 and L2 samples except for fat, which was significantly lower (*p* < 0.001) for the L2 vat milk with a high BCS load. Consequently, the fat/protein ratio of L2 vat milk was also lower than L1 (*p* < 0.01, Table 3). Natural creaming involves most of the bacteria, somatic cells, and spores physically and/or chemically interacting with the fat globule clusters and, surfacing at the top of the milk, then being removed with the cream [27]. In our study, milk creaming failed to minimize the difference in spore load between L1 and L2 milks, although milk defatting seemed improved where higher levels of spore-formers were present. In fact, the amount (kg) of cream raised and recovered was higher for L2 milk (*p* < 0.001), thus resulting in lower fat content in L2 skimmed milk samples (Table 2). Specific studies will be needed to explain this phenomenon.

Table 4 reports values of salt, pH, and the main organic acids (lactic, acetic, butyric, and propionic) detected both in a slice of L1 and L2 cheeses representative of the whole cheese (i.e., sampled along the radius of the wheel, from the outer to the innermost layer, according to Tidona et al. [9]) and their outermost and innermost parts. Expectedly, the percentage of fat and protein on cheese dry matter (DM) and the fat/protein ratio were also significantly lower in the L2 mature cheese samples, while no differences were found between L1 and L2 regarding DM, ash, and salt. The pH values and concentrations of organic acids determined for the whole cheese wheel did not show significant differences between L1 and L2 cheeses (Table 4). In hard, long-ripened cheeses, the butyric fermentation carried out by the butyric clostridia is generally to the detriment of the residual lactic acid present in the product [1]. In our samples, this was evidenced by investigating the composition in different cheese layers. In the innermost parts of L2 samples, mainly affected by LB defect, the average amount of butyric acid and the pH were significantly higher, and the lactic acid content lower, than L1 cheese samples, while no differences between L1 and L2 outer parts of cheese slices were observed (Table 4). Interestingly, the salt content between the whole slice and its external and internal parts did not differ between L1 and L2 cheeses, thus excluding, among the possible causes of the LB defect, a different penetration of NaCl between the two series of cheeses (Table 4). However, the salt content of the innermost parts, both for L1 and L2 cheeses, was significantly lower than the outer parts (Appendix A). In the core of the cheese, the low NaCl concentration and the high moisture support the germination of spores into vegetative cells, thus favoring butyric acid fermentation, further facilitated in an anaerobic environment [8]. In large-sized cheeses such as Grana and similar, very low redox values are established in the innermost parts of the cheese, thus explaining why the LB defect generally takes place there (Appendix A). In addition to the presence of BCS, W-L1 cheeses (e.g., L1 cheeses produced in the warm season that showed texture defects) (Table 1; Appendix A) contained, in their innermost part, butyric acid at concentrations >0.3 g kg^−1^ of DM cheese (Table 4), confirming that an LB defect probably also took place in this subset of cheeses. An amount of butyric acid greater than 100–200 mg kg^−1^ is considered indicative of butyric acid fermentation and the critical threshold for distinguishing LB-affected from non-LB-affected cheeses [28]. In Grana Padano cheeses with LB defect, Cocolin et al. [29] found levels of butyric acid ranging from undetectable levels to 1490 mg kg^−1^ of cheese DM, determined on cheese samples not referred to a specific layer. The presence and the amount of butyric acid, however, also depend on factors such as cheese variety and cheese age, but it was also found to be clostridia species- and strain-related [3]. 

A further analysis of the data, considering the subset of the cheesemaking trials (W-L1 and W-L2), carried out in the warm season, confirmed the presence of a climate effect (C vs. W) that, often in addition to the milk effect (L1 vs. L2), may contribute to explaining the significant differences in chemical composition and some technological parameters between the two series of cheeses. In the warm season, the environmental temperature at the beginning and the end of creaming was expectedly higher than the cold season, with a shorter creaming time; moreover, lower final pH and titratable acidity values were scored in milk for W-L1 and W-L2 trials (Table 2). Table 3 reports milk (L1 vs. L2) and climate (C vs. W) effects on vat milk composition, technological parameters, and cheese whey and curd composition recorded for each cheesemaking trial. Both pH and titratable acidity (°SH) values of the whey starter were affected by an interaction between milk (L1 vs. L2) and season (C vs. W). pH and titratable acidity of vat milk during whey starter addition, and titratable acidity of vat milk at rennet time addition were also influenced by a climate effect. Differently from L2 samples, a significantly higher titratable acidity in the cheese whey after cooking and at curd extraction were recorded in L1 during the cold period (C-L1 vs. W-L1, Table 3). Compared to cheese produced in the cold season, a shorter time of brine salting and a lower yield both before and after salting characterized cheese produced in the warm season (C-L1 and C-L2 vs. W-L1 and W-L2; Table 3). Overall, the data relating to the subset of L1 cheeses produced in the warm season (W-L1), which have shown a tendency to develop LB, seem to suggest that a series of technological, compositional, and environmental, season-related factors may determine the onset of the defect even with BCS levels, in both milk and cheeses, significantly lower than the corresponding subset W-L2 (Table 1).

All the twelve GP cheese samples were analyzed using metabarcoding. Overall, 1,100,008 reads were sequenced, with 84,616 reads per sample on average (range 38,118–270,833). A total of 187 OTUs, having >5 reads were identified, of which 46 were further split into 17 dominant (≥1% total reads) and 29 subdominant (0.1–1% total reads) taxa (data not shown). Metabarcoding data were analyzed to evaluate the relative abundance of the dominant bacterial species (17 taxa) found in L1 and L2 Grana-type cheese samples (Figure 1). *Lacticaseibacillus* (*Lcb.*) *rhamnosus*, *Lactobacillus* (*Lb.*) *helveticus*, *Streptococcus* (*S.*) *thermophilus*, *Lb. delbrueckii*, and *Limosilactobacillus* (*Lim.*) *fermentum* were the five species present in the highest percentages in most cheese samples, thus confirming previous studies [7]. Moreover, a high abundance of *Lactococcus* spp. in L1 cheeses and the presence of *Cl. tyrobutyricum* within the top dominant taxa in L2 cheese samples was shown (Figure 1). Alpha diversity showed a similar richness (d.f. = 1, F = 0.78, *p* = 0.39) but a statistically different Shannon index (d.f. = 1, F = 9.70, *p* = 0.01) between L1 and L2 cheeses. Then, the microbial composition of the milk produced from the two couples of farms with different BCS titers significantly influenced the bacterial evenness (Appendix A). It can therefore be stated that it is not the overall structure of the microbial community that differentiates L1 from L2 cheeses but rather the relative distribution of the species between them. Specifically, metagenomic data confirmed that L2 differs from L1 mainly in the more abundant presence of *Cl. tyrobutyricum*, also present in L1, although to a lesser extent than in L2 (Figure 1), which once more justifies the onset of LB in L2 cheeses.

The LB defect in hard and long-ripened cheeses is still widespread and causes a depreciation of the product. To date, experimental evidence shows that LB could be counteracted through a correct management of the process steps that prevent BCS germination (curd acidification, salting, correct maturation) during cheese production and using lysozyme. However, all these obstacles can be ineffective if the number of spores in the milk is too high [4,16,23]. Various measures regarding farm and milking hygiene and feed preparation have been implemented so far to minimize the risk of contamination of raw milk with butyric clostridial spores (BCS). Silage is the main source of BCS in milk. Careful animal cleaning and good milking practices are therefore essential to maintain low levels of spore contamination in bulk tank milk. To effectively minimize the number of *Clostridium* spp. and *Paenibacillus* spp. spores, these practices should however be foregone by correct silage management and cleaning during total mixed ration preparation [16,23,30,31].

However, reducing high numbers of BCS in the milk can prove to be a very difficult task. Considering that thermal treatments are ineffective, they can be eliminated using either physical (bactofugation, microfiltration) or enzymatic (lysozyme) methods. Lysozyme is effective as it is specifically active against *Cl. tyrobutyricum*, which is the species most involved in the LB defect. On the other hand, its action is ineffective with too high levels of spores in the milk (>5000–10,000 L^−1^) or in case of low susceptibility to the enzyme (frequent in *Cl. sporogenes*) [1]. Therefore, its addition should be foreseen in the context of an “obstacle” technology, which begins from an excellent bacteriological quality of milk, combined with tools that may contrast BCS development during cheese processing and ripening. Stimulated by health (individual allergenicity to egg proteins) and commercial image (e.g., need to offer the consumer clean-label products) needs, the search for alternative methods to lysozyme (e.g., protective cultures) is assuming growing interest even if, at present, very few practical applications are available.

## 4. Conclusions

The results of this work confirm that a low number of spores in milk can contribute to obtaining LB-free Grana-type hard cheeses without lysozyme if the other variables that help in limiting their development (starter activity, salting, ripening) are correctly applied during dairy processing. This evidence, while logical from a theoretical point of view, has rarely had direct confirmation in experimental cheesemaking. From our data, it is not possible to define the lowest number of spores in milk that would allow not using lysozyme as an anti-clostridial agent. Indeed, the LB defect strongly depends on technology and the qualitative–quantitative ratios in species and strains of spore-forming microbiota present in milk, increasingly able to adapt to stress to which it is subjected during dairy processing and, regardless of the number of spores, also from seasonal effects. Our data suggest that a level of <200 BCS L^−1^ vat milk could be reasonable to limit the onset of LB in Grana cheese. However, our work also underlines that, in the absence of lysozyme, this BCS level may not be sufficient to avoid LB in cheeses produced in the warm season. Future research may be carried out to study the factors that, even in the presence of a limited number of spores and in the absence of lysozyme, may contribute to controlling the onset of the LB defect, especially in cheeses produced during the hot season.

## Figures and Tables

**Figure 1 foods-12-01880-f001:**
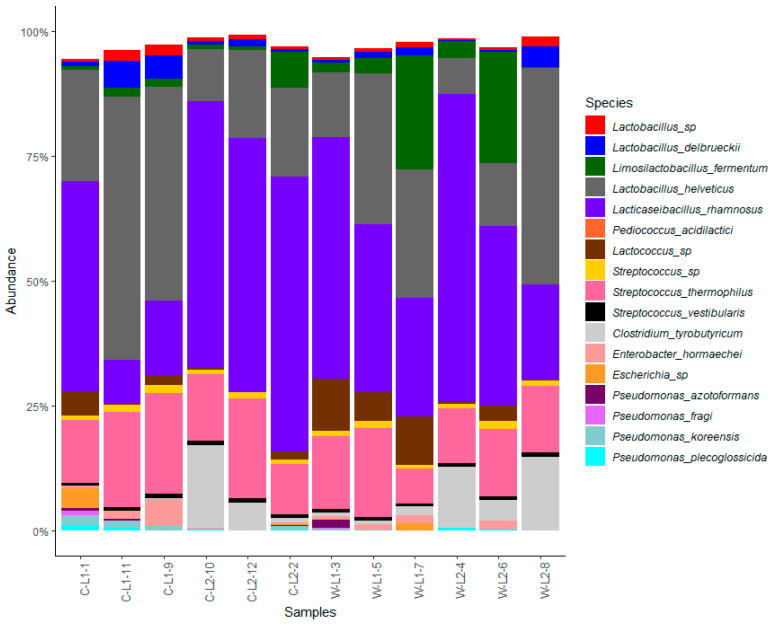
Average values of relative abundance of the 12 dominant taxa retrieved from L1 and L2 Grana-type cheese samples produced in cold (C-L1-1/9/11 and C-L2-2/10/12) and warm (W-L1-3/5/7 and W-L2-2/4/8) periods, respectively.

**Table 1 foods-12-01880-t001:** Anaerobic spore-forming bacteria and butyric clostridia spores count in vat milk used for cheesemaking and in the Grana-type cheeses obtained. Detection of butyric clostridia species using multiplex-PCR on bacterial DNA extracted from MPN-positive tubes.

Cheesemaking Trials			Vat Milk						Cheese					
Milk Quality ^1^	Climatic Period ^2^	Trial nr	ANSB ^3^	BCS ^3^	Butyric Clostridia Species ^4^				ANSB	BCS	Butyric Clostridia Species			
			log MPN L^−1^		TYR	BUT	BEJ	SPOR	log MPN g^−1^		TYR	BUT	BEJ	SPOR
L1	C	1	3.18	3.18	-	+	-	-	−0.32	−0.32	-	-	-	-
L1	W	3	4.66	1.68	-	-	-	-	1.63	−0.32	-	-	-	-
L1	W	5	2.56	1.68	-	-	-	-	0.56	0.56	+	-	-	-
L1	W	7	3.63	1.68	-	-	-	-	0.96	0.96	+	-	-	-
L1	C	9	3.63	1.68	-	-	-	-	0.96	0.96	-	-	-	+
L1	C	11	3.36	1.68	-	-	-	-	0.56	−0.32	-	-	-	-
Average			3.50 ^A^	1.93 ^A^					0.73 ^A^	0.25 ^A^				
SD			0.69	0.61					0.65	0.65				
L2	C	2	3.63	3.36	+	-	-	-	1.36	1.36	+	-	-	-
L2	W	4	2.96	1.68	-	-	-	-	4.66	4.66	+	-	-	-
L2	W	6	3.63	3.36	+	-	-	+	2.38	2.38	+	-	-	-
L2	W	8	2.96	2.96	+	-	-	+	5.04	5.04	+	-	-	-
L2	C	10	3.63	3.63	+	+	-	-	4.66	4.38	+	-	-	-
L2	C	12	3.18	2.96	+	-	-	-	3.38	3.38	+	-	-	-
average			3.33 ^A^	2.99 ^B^*					3.58 ^B^**	3.53 ^B^***				
SD			0.34	0.69					1.48	1.44				

^1^ Milk from farms recognized for low (L1) or high (L2) content of butyric clostridia spores. ^2^ Climate of the period when the cheesemaking was carried out: warm (W), cold (C). ^3^ ANSB: anaerobic spore-forming bacteria; BCS: butyric clostridia spores. ^4^ +: presence; -: absence; TYR: *Cl. tyrobutyricum*; BUT: *Cl. butyricum*; BEJ: *Cl. bejerinckii*; SPOR: *Cl. sporogenes*. ^A,B^ Average values in column with different capital letters are different. Significance (*p*) codes: *** = <0.001; ** = <0.01; * = <0.05.

**Table 2 foods-12-01880-t002:** Least square means, standard errors, and statistical significance for the variables of the creaming process (bulk milk composition, creaming parameters, cream composition) recorded for each cheesemaking trial (observation, *n* = 24). Subset of cheesemaking trials were identified according to the microbiological quality of milk (based on butyric clostridia spore loads. i.e., low, L1 vs. high, L2), and to the microclimate at the time of milk processing (warm, W vs. cool, C).

Item								
Milk quality ^1^	L1		L2		Standard error	Milk ^3^ effect (L1 vs. L2)	Climate ^3^ effect (C vs. H)	Interaction
Climatic period ^2^	C-L1	W-L1	C-L2	W-L2				
Cheesemaking trials (nr)	*n* = 6	*n* = 6	*n* = 6	*n* = 6				
Bulk milk (composition)								
Fat (%)	4.01	3.86	3.90	3.78	0.0219	**	***	-
Protein (%)	3.37	3.26	3.51	3.35	0.0328	**	**	-
Fat-to-protein ratio	1.19	1.18	1.11	1.13	0.0120	***	-	-
Dry matter (%)	13.35	13.10	13.37	13.05	0.0385	-	***	-
Milk creaming process								
Starting parameters:								
Environment temperature (°C)	19.43	22.20	21.43	22.70	0.5522	-	**	-
Milk temperature (°C)	10.47	9.78	10.35	10.22	0.2075	-	-	-
Milk pH	6.71	6.64	6.74	6.70	0.0157	**	*	-
Milk titratable acidity (°SH)	3.30	3.27	3.27	3.10	0.0522	-	-	-
Final parameters:								
Environment temperature (°C)	19.73	23.43	21.28	23.15	0.5721	-	***	-
Milk temperature (°C)	14.35	15.25	15.57	15.65	0.5275	-	-	-
Milk pH	6.71	6.67	6.75	6.67	0.0201	-	*	-
Milk titratable acidity (°SH)	3.37	3.23	3.40	3.17	0.0262	-	***	-
Creaming, duration (min)	393.67	362.00	419.00	366.33	8.7568	-	***	-
Separated cream, quantity (kg)	32.19	34.37	38.92	41.42	1.2410	***	-	-
Cream (composition)								
Fat (%)	21.60	20.93	20.81	19.96	0.4157	*	-	-
Protein (%)	2.85	2.78	2.90	2.85	0.0260	*	*	-
Dry matter (%)	29.36	28.60	28.63	27.71	0.3994	*	*	-

^1^ Milk from farms recognized for low (L1) or high (L2) content of butyric clostridia spores. ^2^ Climate of the period when the cheesemaking was carried out: warm (W), cool (C). ^3^ Significance (*p*) codes: *** = <0.001; ** = <0.01; * = <0.05; - = >0.05.

**Table 3 foods-12-01880-t003:** Least square means, standard error, and statistical significance for the variables of the cheesemaking process (vat milk composition, technological parameters, cheese whey, and curd composition) recorded for each cheesemaking trials (total observation, *n* = 24). Subset of cheesemaking trials were identified according to the microbiological quality of milk (based on butyric clostridia spore loads, i.e., low, L1 vs. high, L2), and to the microclimate at the time of milk processing (warm, W vs. cool, C).

Item								
Milk quality ^1^	L1		L2		Standard error	Milk Effect ^3^ (L1 vs. L2)	Climate Effect ^3^ (C vs. W)	Interaction
Climatic period ^2^	C-L1	W-L1	C-L2	W-L2				
Cheesemaking trials (nr)	*n* = 6	*n* = 6	*n* = 6	*n* = 6				
Vat milk								
Temperature (°C)	14.80	14.00	13.98	14.08	0.3024	-	-	-
pH	6.69	6.66	6.72	6.66	0.0194	-	-	-
Titratable acidity (°SH/50 mL)	3.43	3.30	3.37	3.14	0.0315	*	***	-
Fat (%)	2.90	2.71	2.62	2.49	0.0472	***	**	-
Protein (%)	3.41	3.29	3.53	3.38	0.0362	**	**	-
Fat-to-casein ratio	1.09	1.06	0.94	0.95	0.0211	***	-	-
Dry matter (%)	12.35	12.07	12.19	11.90	0.0600	**	***	-
Technological parameters								
Whey starter culture, pH	3.41	3.29	3.35	3.37	0.0279	-	-	*
Whey starter culture, acidity (°SH)	28.67	28.03	28.07	29.33	0.2532	-	-	**
Whey starter culture, temperature (°C)	20.60	22.60	21.10	23.47	0.2597	*	***	-
Whey starter culture, quantity (g)	1500.00	1500.00	1733.33	1516.67	57.0283	*	-	-
Vat milk after whey starter addition, pH	6.44	6.42	6.45	6.42	0.0055	-	***	-
Vat milk after whey starter addition, acidity (°SH)	4.22	4.00	4.20	3.97	0.0506	-	***	-
Vat milk after whey starter addition, temperature (°C)	20.30	21.57	21.62	21.57	0.4149	-	-	-
Vat milk at time of rennet addition, pH	6.40	6.41	6.41	6.41	0.0039	-	-	-
Vat milk at time of rennet addition, acidity (°SH)	4.27	4.05	4.20	3.98	0.0444	-	***	-
Vat milk at time of rennet addition, temperature (°C)	33.38	33.25	33.15	33.27	0.0399	*	-	**
Coagulation time (min)	10.56	10.99	10.84	10.93	0.3351	-	-	-
Manual cutting of coagulum, time (sec)	41.17	57.33	46.33	48.00	8.1320	-	-	-
Mechanical cutting of coagulum, time (sec)	76.33	62.33	78.00	82.83	5.6151	-	-	-
Curd cooking, temperature (°C)	53.30	53.45	53.42	53.70	0.0621	**	**	-
Curd cooking, time (min)	5.09	5.33	5.09	5.12	0.1059	-	-	-
Curd under whey, time (min)	67.67	67.67	66.83	71.17	1.9713	-	-	-
Cheese whey after cooking, pH	6.30	6.32	6.31	6.30	0.0062	-	-	*
Cheese whey after cooking, acidity (°SH)	2.93	2.70	2.77	2.75	0.0514	-	*	*
Cheese whey after cooking, temperature (°C)	53.22	53.37	53.28	53.72	0.1648	-	-	-
Cheese whey after curd extraction, pH	6.17	6.09	6.10	6.09	0.0461	-	-	-
Cheese whey after curd extraction, acidity (°SH)	3.23	3.07	3.03	3.02	0.0569	*	-	-
Cheese whey after curd extraction, temperature (°C)	49.88	50.83	50.50	51.20	0.3740	-	*	-
Curd drainage, time (min)	42.67	31.33	33.67	32.50	3.5538	-	-	-
Curd molding, first turning time (min)	147.33	174.67	157.17	175.83	13.3804	-	-	-
Curd molding, second turning time (min)	170.00	170.00	175.00	172.50	4.3661	-	-	-
Curd, brine salting time (d)	23.33	20.50	23.33	20.50	1.1643	-	*	-
Curd yield before salting (%)	8.93	8.52	8.85	8.41	0.1058	-	***	-
Curd yield after salting (%)	8.64	8.26	8.52	8.15	0.1061	-	***	-
Residual cheese whey (composition)								
Fat (%)	0.56	0.55	0.51	0.41	0.0516	-	-	-
Protein (%)	0.95	0.91	0.99	0.96	0.0116	***	*	-
Dry matter (%)	7.75	7.70	7.66	7.58	0.0603	-	-	-
Curd (composition)								
pH	5.41	5.43	5.28	5.29	0.0745	-	-	-
Fat (%)	26.76	26.16	24.12	24.41	0.3930	***	-	-
Protein (%)	31.04	30.62	32.24	31.84	0.4467	*	-	-
Dry matter (%)	60.94	59.66	59.02	59.41	0.5197	-	-	-
Fat-to-DM ratio	0.44	0.44	0.41	0.41	0.0060	***	-	-
Protein-to-DM ratio	0.51	0.51	0.55	0.54	0.0050	***	-	-
Fat-to-protein ratio	0.87	0.85	0.75	0.77	0.0184	***	-	-

^1^ Milk from farms recognized for low (L1) or high (L2) content of butyric clostridia spores. ^2^ Climate of the period when the cheesemaking was carried out: warm (W), cool (C). ^3^ Significance (*p*) codes: *** = <0.001; ** = <0.01; * = <0.05; - = >0.05.

**Table 4 foods-12-01880-t004:** Least square means, standard error, and statistical significance for the variables of cheeses produced in the different cheesemaking trials (total observation, *n* = 12). Subset of cheesemaking trials were identified according to the microbiological quality of milk (based on butyric clostridia spore loads. i.e., low, L1 vs. high, L2), and to the microclimate at the time of milk processing (warm, W vs. cold, C).

Item								
Milk quality ^1^	L1		L2		Standard error	Milk effect ^3^	Climate effect ^3^	Interaction
Climatic period ^2^	C-L1	W-L1	C-L2	W-L2				
Cheesemaking trials (nr)	*n* = 3	*n* = 3	*n* = 3	*n* = 3		(L1 vs. L2)	(C vs. W)	
Cheese								
Ripening time (months)	8.83	10.67	8.33	10.50	1.3307	-	-	-
Dry matter (%)	66.56	67.04	65.18	66.85	0.4738	-	-	-
Ash (%)	4.58	4.59	4.83	4.61	0.1353	-	-	-
Fat (%)	29.08	29.04	26.45	27.91	0.5081	**	-	-
Protein (%)	31.06	31.57	32.10	32.52	0.3122	*	-	-
Fat-to-DM ratio	0.44	0.43	0.41	0.42	0.0060	**	-	-
Protein-to-DM ratio	0.47	0.47	0.49	0.49	0.0054	**	-	-
Fat-to-protein ratio	0.94	0.92	0.82	0.86	0.0226	**	-	-
Salt (%)	1.70	1.62	1.93	1.62	0.1390	-	-	-
Salt, outer zone (%)	1.91	1.81	2.08	1.82	0.1178	-	-	-
Salt, inner zone (%)	1.23	1.29	1.26	1.16	0.2011	-	-	-
pH	5.30	5.38	5.37	5.51	0.0418	-	*	-
pH, outer zone	5.32	5.39	5.31	5.42	0.0255	-	**	-
pH, inner zone	5.32	5.42	5.55	5.73	0.0961	*	-	-
Lactic acid (g-to-kg DM)	25.18	24.33	24.22	22.82	0.9464	-	-	-
Lactic acid, outer zone (g-to-kg DM)	23.71	24.08	24.58	23.53	0.8231	-	-	-
Lactic acid, inner zone (g-to-kg DM)	25.03	22.96	18.04	15.32	31.407	*	-	-
Butyric acid (g-to-kg DM)	0.00	0.00	0.69	0.93	0.3563	-	-	-
Butyric acid, outer zone (g-to-kg DM)	0.00	0.00	0.00	0.00	0.0000	-	-	-
Butyric acid, inner zone (g-to-kg DM)	0.00	0.50	3.79	4.13	12.950	*	-	-
Propionic acid (g-to-kg DM)	0.00	0.00	0.10	0.08	0.0632	-	-	-
Propionic acid, outer zone (g-to-kg DM)	0.00	0.00	0.00	0.00	0.0000	-	-	-
Propionic acid, inner zone (g-to-kg DM)	0.00	0.00	0.58	0.74	0.3013	-	-	-
Acetic acid (g-to-kg DM)	1.61	2.08	1.87	2.05	0.1255	-	*	-
Acetic acid, outer zone (g-to-kg DM)	1.52	1.99	1.79	1.98	0.0916	-	**	-
Acetic acid, inner zone (g-to-kg DM)	1.49	1.98	1.44	1.70	0.2564	-	-	-
Succinic acid (g-to-kg DM)	0.92	0.94	0.89	0.87	0.0523	-	-	-
Pyroglutamic acid (g-to-kg DM)	3.50	4.01	3.27	4.02	0.5037	-	-	-

^1^ Milk from farms recognized for low (L1) or high (L2) content of butyric clostridia spores. ^2^ Climate of the period when the cheesemaking was carried out: hot (H), cool (C). ^3^ Significance (*p*) codes: ** = <0.01; * = <0.05; - = >0.05.

## Data Availability

Data is contained within the article or Appendix A.

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
