# Peer review of "Low-Level Clostridial Spores’ Milk to Limit the Onset of Late Blowing Defect in Lysozyme-Free, Grana-Type Cheese"

_foods, 2023, doi:10.3390/foods12091880_

Round 1
Reviewer 1 Report
The manuscript entitled ‘ Grana cheese, butyric clostridia, and late blowing: is it possible to manufacture it without lysozyme?’ is an interesting study however, there is scope for improvement. Kindly address the following comments carefully:
General comment:
Language needs improvement.
Abstract
The abstract could be more specific and short.
Line 11: What does PDO mean?
Line 11: Give a few examples of similar cheeses.
Line 16: Give once the full form of MPN?
Keywords: Try to limit the keywords up to 6 and avoid the words from title itself.
Introduction
Enrich the section with information on types of milk (L1 and L2) and their use in GP. Moreover, give a few lines as future prospects of the present study in the end.
Materials and methods
Give separate sub section on chemicals and reagents used.
Line 79: Provide the quantity of procured milk (L1 & L2) used for the study.
Line 80: Name the dairy cooperative.
Line 87: ‘six with milk L1 and 6 with …’ Keep a single format i.e., either ‘6’ or ‘six’.
Line 91: It is not clear which milk type was used in which month. Either write ‘respectively’ in the endo of the sentence or clarify it.
Line 100: Provide the size (dimensions) of slice.
Line 101: What was the amount of sample?
Sub-section chemical analysis: This section is not fully explained. Better to provide detail of most of the methods/analyses. What were the conditions of HPLC, used standards, and column details?
Line 133: How an Excel spreadsheet was used to estimate the microbiological parameters? Explain.
What about the texture and sensory attributes of the prepared cheeses? These are the important parameters.
Results and discussion
Discussion part can be improved.
Author Response
Reviewer 1 - Comments and Suggestions for Authors - foods-2341523
The manuscript entitled ‘Grana cheese, butyric clostridia, and late blowing: is it possible to manufacture it without lysozyme?’ is an interesting study however, there is scope for improvement. Kindly address the following comments carefully:
General comment:
Language needs improvement. After revision it has been improved.
Abstract
The abstract could be more specific and shorter. It was partially shortened.
Line 11: What does PDO mean? Protected Designation of Origin.
Line 11: Give a few examples of similar cheeses: other hard and semi-hard, long ripened cheeses. Done, see lines 11-12 of the revised text.
Line 16: Give once the full form of MPN? Most Probable Number.
Keywords: Try to limit the keywords up to 6 and avoid the words from title itself. Done
Introduction
Enrich the section with information on types of milk (L1 and L2) and their use in GP. Moreover, give a few lines as prospects of the present study in the end. Prospects are usually put at the end of the paper. Therefore, we added a sentence at the end of the Conclusions (see lines 422-425 of the revised text).
Materials and methods
Give separate sub section on chemicals and reagents used. As we did not used many chemicals and reagents and most of the methods were described in the literature and referee to official protocols, we though not to add a separate section on it.
Line 79: Provide the quantity of procured milk (L1 & L2) used for the study. This is already reported later in the text (lines 99-102 of the revised text).
Line 80: Name the dairy cooperative. See modified text, line 88 of the revised text.
Line 87: Done (line 96 of the revised text).
Line 91: It is not clear which milk type was used in which month. Either write ‘respectively’ in the end of the sentence or clarify it. The section has been rephrased, see lines 99-102 of the revised text.
Line 100: Provide the size (dimensions) of slice. Done, see line 108 of the revised text.
Line 101: What was the amount of sample? Done, see line 108 of the revised text.
Line 106: Sub-section chemical analysis: This section is not fully explained. Better to provide detail of most of the methods/analyses. Please refer to the reference list: most of the methods are official methods.
Line 113: What were the conditions of HPLC, used standards, and column details? Done, see lines 121-126 of the revised text.
Line 133: How an Excel spreadsheet was used to estimate the microbiological parameters? Explain. The explanation can be found in the cites reference of Jarvis et al. However, the sentence has partially been rewritten (see lines 146-148 of the revised text).
What about the texture and sensory attributes of the prepared cheeses? These are the important parameters. We did not carry out sensory evaluation of cheese. Our aim was just to score the presence of late blowing, so mostly structural defects. To this aim, we showed pictures of cheeses as supplementary figures.
Results and discussion
Discussion part can be improved. We did it.
Reviewer 2 Report
The manuscript with Number FOODS 2341523 is interestingly focused on the possibility to manufacture Grana cheese produced with vat milk contaminated with clostridial butyric spores and in different seasons (cold and warm). The study was carried out with cheeses produced without lysozyme addition, while lysozyme is used to avoid late blowing.
The authors should consider the following reflexions:
The Title is “Grana cheese, butyric clostridia, and late blowing: is it possible to manufacture it without lysozyme?”
The authors could think about the title since that question has not been clearly answer in results (and therefore neither in abstract nor in conclusions).
In the Abstract.
The last sentence (lines 35-38) says: “It was not possible to establish a specific level of spores in milk that could be safe, as it strongly depends on many concomitant factors including, according to our data and independently of the number of clostridial butyric spores, also from seasonal variations”.
The authors should think about the aim of the study since it is “avoiding late blowing in Grana cheese produced by clostridial butyric spores without lysozyme addition”. And for that it is necessary to establish a specific level of spores in milk that could be safe and, besides, because the seasonal variations have been studied.
In Materials and methods
Line 79: Four farms. How many were L1 or L2 farms?
Line 123: MPN, should be explained/written the first time as “more probable number”.
In Results and discussion.
Line 251-254. The number of the table should be mentioned. Also, the data from the table should be included in the text for better understanding of the results.
Line 224-225. When referred to vat is Table 3, or bulk is Table 2.
Line 261. Should it be Table 2 instead of 3? Also, the data from the table should be included in the text for better understanding of the results.
Line 286. The data from the table should be included in the text for better understanding of the results.
Line 321. However, the temperatures of milk were not different, is it not important?
Line 322. The data of pH from the table should be included in the text for better understanding of the results. Do the authors think that the pH differences could impact on the microbial growth or late blowing?
Lines370-375. “Silage…31)” could be in introduction but clearly not in here.
On my opinion Table 4 does not give important information related to the aim or focus of this study.
Have the authors the results about which cheeses (C-L1, W-L1, C-L2, W-L2) presented problems of late blowing? It is necessary to be included in the manuscript.
In Conclusions.
Lines 399-401. Being it the focus of the study, do the authors think that is it possible to conclude “From our data it is not conceivable to establish a specific limit level (or range) of spores in milk that could be 'safe', especially if one wished to avoid the use of lysozyme as an anti-clostridia agent”?
Lines 401-404. References should not be included in conclusions.
Line 407. Can be the results of this study be applied in similar cheeses? Why?
Line 405-407. Then, is the final conclusion “As a rule, our data suggests that a level of < 200 BCS L-1 vat milk could reasonably prevent, or limit, the onset of LB in Grana-like”?
Line 407-408. Is it also a conclusion “Our work also confirms that, in the absence of lysozyme, this BCS level may not be sufficient to avoid LB in cheeses produced in the warm season”? It is opposite to the conclusion in lines 405-407.
Lines 409-410. That is not a conclusion of this study.
Author Response
Reviewer 2 - Comments and Suggestions for Authors - foods-2341523
The manuscript with Number FOODS 2341523 is interestingly focused on the possibility to manufacture Grana cheese produced with vat milk contaminated with clostridial butyric spores and in different seasons (cold and warm). The study was carried out with cheeses produced without lysozyme addition, while lysozyme is used to avoid late blowing. Thank you for your positive comments.
The authors should consider the following reflexions:
The Title is “Grana cheese, butyric clostridia, and late blowing: is it possible to manufacture it without lysozyme?” The authors could think about the title since that question has not been clearly answer in results (and therefore neither in abstract nor in conclusions). The title has been changes according to your observation as follows: ‘Low-level clostridial spores’ milk to limit the onset of late blowing defect in lysozyme-free Grana cheese’.
In the Abstract.
The last sentence (lines 35-38) says: “It was not possible to establish a specific level of spores in milk that could be safe, as it strongly depends on many concomitant factors including, according to our data and independently of the number of clostridial butyric spores, also from seasonal variations”.
The authors should think about the aim of the study since it is “avoiding late blowing in Grana cheese produced by clostridial butyric spores without lysozyme addition”. And for that it is necessary to establish a specific level of spores in milk that could be safe and, besides, because the seasonal variations have been studied.
You are right; indeed, a sentence was already included in the conclusions, but we forgot to revise the abstract accordingly. So, we rephrased (lines 33-37 of the revised text).
In Materials and methods
Line 79: Four farms. How many were L1 or L2 farms? The sentence was rephrased (see lines 91-93 of the revised text.)
Line 123: MPN, should be explained/written the first time as “Most Probable Number”. Done.
In Results and discussion.
Line 251-254. The number of the table should be mentioned. Also, the data from the table should be included in the text for better understanding of the results. The sentence has been rephrased (see lines 267-270 of the revised text).
Line 254-255. When referred to vat is Table 3, or bulk is Table 2. Ok, done (see above rephrasing).
Line 261. Should it be Table 2 instead of 3? Yes.
Also, the data from the table should be included in the text for better understanding of the results. We would like to leave them in the table to avoid repetitions and too long paper text. Line 286. The data from the table should be included in the text for better understanding of the results. We would like to leave them in the table to avoid repetitions and too long paper text.
Line 321. However, the temperatures of milk were not different, is it not important? That’s true, but this does not surprise us as the creaming time of a large mass of milk in the creaming trays does not significantly affect the temperature of the milk itself, cold stored before creaming. However, we did not carry out a specific temperature effect on the overall creaming and further technological steps. When we used the term ‘climate effect’ we meant an overall, multifactorial, incidence of the season on the chemical and physical-chemical milk composition. That’s why we did not comment this fact.
Line 322. The data of pH from the table should be included in the text for better understanding of the results. We would like to leave them in the table to avoid repetitions and too long paper text.
Do the authors think that the pH differences could impact on the microbial growth or late blowing? We are talking about pH difference in milk, which could be indicator of a different microbial growth/composition in milk, not in cheese.
Lines 370-375. “Silage…31)” could be in introduction but clearly not in here. We agree; the sentence has been removed.
On my opinion Table 4 does not give important information related to the aim or focus of this study. Table 4 shows how cheeses obtained with a high number of spores, compared to products with a low number of spores, are more prone to a late blowing effect, demonstrated by the presence of significant butyric acid in the center of the wheels (milk effect). Furthermore, the statistical processing and the comparison between the two groups of cheeses highlight the season effect (climate effect) on L1 cheeses. Therefore, in our opinion, Table 4 is substantial and necessary to support the discussion of the data.
Have the authors the results about which cheeses (C-L1, W-L1, C-L2, W-L2) presented problems of late blowing? It is necessary to be included in the manuscript. The part of text between lines 300 and 316 of the revised text refers to which cheeses have been affected by the LB defect: that is, all the L2s and only the L1s produced in the hot season. In the same section, reference is made to Figure S1, which shows the images of the cheeses.
In Conclusions.
Lines 399-401. Being it the focus of the study, do the authors think that is it possible to conclude “From our data it is not conceivable to establish a specific limit level (or range) of spores in milk that could be 'safe', especially if one wished to avoid the use of lysozyme as an anti-clostridia agent”? We modified the sentence (see lines 412-414 of the revised text)
Lines 401-404. References should not be included in conclusions. Ok, we deleted.
Line 407. Can be the results of this study be applied in similar cheeses? Why? We hypothesize that results could be extended to cheeses with similar technology and problems (e.g., hard, long ripened cheeses) but we have no direct proofs of it, then we preferred to delete.
Lines 405-407. Then, is the conclusion “As a rule, our data suggests that a level of < 200 BCS L-1 vat milk could reasonably prevent, or limit, the onset of LB in Grana-like”?
Lines 407-408. Is it also a conclusion “Our work also confirms that, in the absence of lysozyme, this BCS level may not be sufficient to avoid LB in cheeses produced in the warm season.”? It is opposite to the conclusion in lines 405-407.
Answers to lines 405-408: we modified the text trying to better explain our message (see lines 417-420 of the revised text).
Lines 409-410. That is not a conclusion of this study. Ok, deleted.
Reviewer 3 Report
foods-2341523-peer-review-v1
The present work is interesting; however, some parts need to be presented, explained better. Authors are experts in cheeses, but not all readers are, and thus is why some mode details needs to be provided in the material and methods. Moreover, in the presentation of the introduction it is better to state the objectives of the research. Based on present way, is giving the impression that authors just want to confirm old idea that primary microbiology quality of the milk is critical for the quality of the cheeses. Well, we all know this. However, if the objective can be better stated, this can show the novelty of the research.
Maybe extended descriptions of the material and methods will be a good idea in improving the paper.
Introduction: Authors in fact suggest and show that already very well-known fact that microbial quality of the milk is the principal argument for production of high-quality cheese. Good manufacturing practices for milk production, including milking, transportation and production are essential for later quality of the cheeses. In this regard, the present paper proves the already existing concept that microbial quality of the prime material (milk) is essential for the quality of the cheese. However, what will be alternative to reduce clostridia and to prevent their negative role in the cheese quality?
Ln2: Please, check if need to be "Grana" or "Grana Padano" or "Grana-type cheeses". As Italians you know better what more appropriate way is to call this cheese.
Ln8: Please, remove on of the "correspondence"
Ln20: Please, correct "-1" to exponential position.
Ln23: Maybe a bit more appropriate term can be used and replace word "tubes".
Ln51-52: Please, full name for all bacterial species needs to be provided when introduced for the first time, even if other species from same genus was already introduced.
Ln79-94: authors have prepared 12 experimental cheeses according to the recommendations for production of Grana Padano cheese - with low and high levels of BCS. All of them without lysozyme. However, it will not be good to have control cheese, where lysozyme was used and by this way to compare results? As well, by provided information is not very clear if these 12 cheeses were produced at ones and then maturated in period April - October, or different cheeses were produced during this period at different time points. Please, this point needs to be explained better. In the abstract even authors compare cheese produced in cold and that produced in warm periods of the year.
Ln98: Why only at one point - after 10 months sample was taken? Any specific reason? What about sampling at any other intermedia time point not been considered in the experimental protocols?
Ln141: Please, centrifugation need to be as "xg", not as "rpm".
Ln142: Can you specify what PCR, targeting what genes were performed. In present way, it is not very clear.
Authors need to describe criteria applied in the analysis of the appearances of the cheese.
Ln202: Please, replace "(" with "["
Multiplex PCR conditions need to be presented in the M&M.
Ln281: Please, correct to (Table 4).
Figure 1: Please, in the legend explain what is 1,11, 3,5,7,9,10,12,2,4,6,8. Maybe group them by cold and warm periods?
Author Response
Reviewer 3 - Comments and Suggestions for Authors - foods-2341523
The present work is interesting; however, some parts need to be presented, explained better. Authors are experts in cheeses, but not all readers are, and thus is why some mode details needs to be provided in the material and methods. Moreover, in the presentation of the introduction it is better to state the objectives of the research. Based on present way, is giving the impression that authors just want to confirm old idea that primary microbiology quality of the milk is critical for the quality of the cheeses. Well, we all know this. However, if the objective can be better stated, this can show the novelty of the research.
Maybe extended descriptions of the material and methods will be a good idea in improving the paper.
Introduction:
Authors in fact suggest and show that already very well-known fact that microbial quality of the milk is the principal argument for production of high-quality cheese. Good manufacturing practices for milk production, including milking, transportation and production are essential for later quality of the cheeses. In this regard, the present paper proves the already existing concept that microbial quality of the prime material (milk) is essential for the quality of the cheese. However, what will be alternative to reduce clostridia and to prevent their negative role in the cheese quality?
This study aimed to verify the cause-effect relationship between the load of BCS in milk and the onset of the LB defect. In this sense, our goal was not so much to start from something as obvious as evident, that is, few spores = good products. If anything, we wanted to have an answer to this axiom on an experimental basis, and in this sense, we also found that there is no optimal level of so-called ‘safety’, as the manifestation of the LB defect is much less evident and above all multifactorial. There is a 'shadow cone' of spore level, let's call it 'intermediate', in which the products may or may not be non-defective. This element has been demonstrated experimentally and, in this sense, we consider it one of the novelties of the work. Furthermore, always experimentally, we verified these hypotheses starting from cheeses produced without lysozyme as a technological adjuvant, evidence that had rarely had so far been verified in dairy practice.
Ln2: Please, check if need to be "Grana" or "Grana Padano" or "Grana-type cheeses". As Italians you know better what more appropriate way is to call this cheese. Thank you, we changed all to ‘Grana-type’.
Ln8: Please, remove on of the "correspondence". Done.
Ln20: Please, correct "-1" to exponential position. Done.
Ln23: Maybe a bit more appropriate term can be used and replace word "tubes". Changed to ‘samples’.
Ln51-52: Please, full name for all bacterial species needs to be provided when introduced for the first time, even if other species from same genus was already introduced. Done.
Ln79-94: authors have prepared 12 experimental cheeses according to the recommendations for production of Grana Padano cheese - with low and high levels of BCS. All of them without lysozyme. However, it will not be good to have control cheese, where lysozyme was used and by this way to compare results? Our aim was to compare low number to high number spore level in milk to experimentally verify that with a low clostridial sporeformers you can o avoid, or limit, the LB defect in mature d cheeses. To make easier the LB inset, we choose not to add lysozyme to both milk types.
As well, by provided information is not very clear if these 12 cheeses were produced at ones and then maturated in period April - October, or different cheeses were produced during this period at different time points. Please, this point needs to be explained better. In the abstract even authors compare cheese produced in cold and that produced in warm periods of the year. Different cheeses were produced during this period at different time points. The sentence has been clarified (see lines 99-102 of the revised text).
Ln98: Why only at one point - after 10 months sample was taken? Any specific reason? What about sampling at any other intermedia time point not been considered in the experimental protocols? Our objective was not to follow the dynamics of the late blowing but to wait a suitable time for the defect to eventually manifest itself. The literature, and our personal experience, indicate that normally the defect occurs after a certain number of months of maturation, time necessary to allow an increase in pH which in turn, again according to the literature, facilitates the germination of the spores.
Ln141: Please, centrifugation need to be as "xg", not as "rpm". Done.
Ln142: Can you specify what PCR, targeting what genes were performed. In present way, it is not very clear. The multiplex was designed of the following genes: colA, collagenase; nifH, nitrogenase iron protein; hydA, hydrogenase; enr, 2-enoate reductase, according to the conditions described in the paper of Cremonesi, et al., Identification of Clostridium beijerinckii, Cl. butyricum, Cl. sporogenes, Cl. tyrobutyricum isolated from silage, raw milk, and hard cheese by a multiplex PCR assay. J. Dairy Res. 2012, 79, 318-323. The amplified genes were reported in the revised text (lines 151-152 of the revised text).
Authors need to describe criteria applied in the analysis of the appearances of the cheese. A sentence was added to the revised text (lines 111-112 of the revised text).
Ln202: Please, replace "(" with "[" Done.
Multiplex PCR conditions need to be presented in the M&M. See answer to your question at line 142 of the original text.
Ln281: Please, correct to (Table 4). Done.
Figure 1: Please, in the legend explain what is 1,11, 3,5,7,9,10,12,2,4,6,8.
Maybe group them by cold and warm periods? Figure and its legend have been updated.
Reviewer 4 Report
Dear Authors,
The manuscript (foods- 2341523) presented for review is very interesting
Authors, Please note and address the following comments:
Keywords: In my opinion, there are too many keywords, without reading the article it is difficult to know what scientific problem this research solves?
Introduction:
The Introduction chapter is the weakest parts of the work. In my opinion, the introduction does not fully demonstrate a gap in the literature. The authors should better point out the technological problem in cheese production than now. In my opinion, the introduction is not fully introductory and explains why this topic maybe of interest to readers, and not just the producers of Grana Padano cheese.
Materials and Methods: This section is well written.
Results and Discussion: The results and discussion are well written.
Limitation: Are there any limitations of this findings?
Limitation section is needed, because if the question (aim of study) could not be fully answered, the authors should answer what are the limitations of this study.
Conclusion
What further research should be conducted to achieve the assumed research goal? What are the practical and theoretical implications of the research?
The current conclusions are quite enigmatic. Conclusions should answer the questions asked.
What are the authors' recommendations for other scientists.
References
References are cited according to journal rules.
Despite my comments, I believe this paper concerns an important area of research in an international context.
Reviewer
Author Response
Reviewer 4 - Comments and Suggestions for Authors - foods-2341523
Dear Authors,
The manuscript presented for review is very interesting. Thank you for your positive comments
Authors, please note and address the following comments:
Keywords:
In my opinion, there are too many keywords, without reading the article it is difficult to know what scientific problem this research solves? Keywords were reduced.
Introduction:
The Introduction chapter is the weakest parts of the work. In my opinion, the introduction does not fully demonstrate a gap in the literature. The authors should better point out the technological problem in cheese production than now. In my opinion, the introduction is not fully introductory and explains why this topic maybe of interest to readers, and not just the producers of Grana Padano cheese. A sentence has been added (see lines 59-67 of the revised manuscript).
Materials and Methods: This section is well written. Thank you.
Results and Discussion: The results and discussion are well written. Thank you.
Limitation:
Are there any limitations of this findings?
Limitation section is needed, because if the question (aim of study) could not be fully answered, the authors should answer what the limitations of this study are.
From our data it is not possible to define the lowest number of spores in milk that would allow not using lysozyme as an anti-clostridial agent. See the modified sentence in the conclusions, lines 412-414 of the revised text.
Conclusion
What further research should be conducted to achieve the assumed research goal? What are the practical and theoretical implications of the research? Please, see the added sentence, lines 417-420 of the revised text.
The current conclusions are quite enigmatic. Conclusions should answer the questions asked.
What are the authors' recommendations for other scientists. Please, see the added sentence, lines 420-423.
References
References are cited according to journal rules.
Despite my comments, I believe this paper concerns an important area of research in an international context. Thank you.
Round 2
Reviewer 1 Report
The authors did not attempt the queries seriously. Many of the important comments have been neglected or superficially addressed.
Author Response
Reviewer 1 - Comments and Suggestions for Authors - foods-2341523-R1
The authors did not attempt the queries seriously. Many of the important comments have been neglected or superficially addressed. We accept your criticism, we tried to further improve the text to better answering to your questions: We hope you’ll appreciate.
Abstract
The abstract could be more specific and shorter. Wherever possible, it was partially shortened.
Line 11: What does PDO mean? Protected Designation of Origin. Done
Line 11: Give a few examples of similar cheeses: other hard, long ripened cheeses such as Provolone, Comté and similar cheeses. See modified sentence, lines 11-12 of the revised text.
Line 16: Give once the full form of MPN? Most Probable Number. Done. See line 17 of the revised text.
Keywords: Try to limit the keywords up to 6 and avoid the words from title itself. Done: they now are ‘Grana Padano cheese; cheese ripening; cheese spoilage; cheese microbiota; anaerobic sporeforming bacteria, milk quality’. See lines 40-41 of the revised text.
Introduction
Enrich the section with information on types of milk (L1 and L2) and their use in GP. Right, we added the sentence at lines 81-82 of the revised text.
Moreover, give a few lines as prospects of the present study in the end. We added a sentence here (lines 86-89) and another at the end of the Conclusions (see lines 422-425 of the revised text).
Materials and methods
Give separate sub section on chemicals and reagents used. As we did not used many chemicals and reagents and most of the methods were described in the literature and referee to official protocols, we would like not to add a separate section on it.
Line 79: Provide the quantity of procured milk (L1 & L2) used for the study. The quantity was indicated, and the sentence rephrased. Lines 105-109 of the revised text.
Line 80: Name the dairy cooperative. See modified text, line 94 of the revised text.
Line 91: It is not clear which milk type was used in which month. Either write ‘respectively’ in the end of the sentence or clarify it. We agree: the section has been rephrased, see lines 105-109 of the revised text.
Line 100: Provide the size (dimensions) of slice. Done, 20x5 cm, see line 116 of the revised text.
Line 101: What was the amount of sample? Done, 1 kg, see line 116 of the revised text.
Line 106: Sub-section chemical analysis: This section is not fully explained. Better to provide detail of most of the methods/analyses. In answering to this request, please note that most of the methods are official methods.
Line 113: What were the conditions of HPLC, used standards, and column details? Thank you, we did it. See lines 129-134 of the revised text.
Line 133: How an Excel spreadsheet was used to estimate the microbiological parameters? Explain. The sentence has partially been rewritten to clarify (see lines 154-156 of the revised text).
What about the texture and sensory attributes of the prepared cheeses? These are the important parameters. We did not carry out sensory evaluation of cheese. Our aim was just to score the presence of late blowing, so to check matured cheeses for structural defects. We showed pictures of cheeses as supplementary figures. Please refer also to lines 119-120 of the revised text.
Results and discussion:
Discussion part can be improved. According to your comments and those of three other referees (e.g., lines 272-275 of the revised text), it has changed since the first version of the text. We believe that, also thanks to the arbitration, it has concomitantly been improved; on this, we stick to your final judgment and ask, eventually, which specific points still need to be improved.
Reviewer 3 Report
in my opinion authors have corrected the text according to the recommendations and paper can be suggested for publication.
A few editorial adjustments need to be taken into consideration: position of tables, Ln122, 123: Please, adjust positions of indexes and exponential, etc.
Author Response
Reviewer 3 - Comments and Suggestions for Authors - foods-2341523-R1
In my opinion authors have corrected the text according to the recommendations and paper can be suggested for publication.
A few editorial adjustments need to be taken into consideration: position of tables, Ln122, 123: Please, adjust positions of indexes and exponential, etc.
Thank you.
We aligned table indexes, exponentials, and legends.